# Listeria monocytogenes faecal carriage is common and depends on the gut microbiota

Lukas Hafner[1,8], Maxime Pichon [2,3,7,8], Christophe Burucoa[2,3,7], Sophie H. A. Nusser[1], Alexandra Moura[1,4], Marc Garcia-Garcera [5,9] & Marc Lecuit [1,4,6,9✉]

Listeria genus comprises two pathogenic species, L. monocytogenes (Lm) and L. ivanovii, and non-pathogenic species. All can thrive as saprophytes, whereas only pathogenic species cause systemic infections. Identifying Listeria species' respective biotopes is critical to understand the ecological contribution of Listeria virulence. In order to investigate the prevalence and abundance of Listeria species in various sources, we retrieved and analyzed 16S rRNA datasets from MG-RAST metagenomic database. 26% of datasets contain Listeria sensu stricto sequences, and Lm is the most prevalent species, most abundant in soil and host-associated environments, including 5% of human stools. Lm is also detected in 10% of human stool samples from an independent cohort of 900 healthy asymptomatic donors. A specific microbiota signature is associated with Lm faecal carriage, both in humans and experimentally inoculated mice, in which it precedes Lm faecal carriage. These results indicate that Lm faecal carriage is common and depends on the gut microbiota, and suggest that Lm faecal carriage is a crucial yet overlooked consequence of its virulence.

[1] Institut Pasteur, Université de Paris, Inserm U1117, Biology of Infection Unit, 75015 Paris, France. [2] University Hospital of Poitiers, Infectious Agents Department, Bacteriology and Infection Control Laboratory, 86021 Poitiers, France. [3] Université de Poitiers, Faculté de Médecine et de Pharmacie, EA 4331, 86022 Poitiers, France. [4] Institut Pasteur, National Reference Center and WHO Collaborating Center Listeria, 75015 Paris, France. [5] University of Lausanne, Department of Fundamental Microbiology, 1015 Lausanne, Switzerland. [6] Necker-Enfants Malades University Hospital, Division of Infectious Diseases and Tropical Medicine, APHP, Institut Imagine, 75006 Paris, France. [7] Present address: Université de Poitiers, Faculté de Médecine et de Pharmacie, Inserm U1070, 86022 Poitiers, France. [8] These authors contributed equally: Lukas Hafner, Maxime Pichon. [9] These authors jointly supervised this work: Marc Garcia-Garcera, Marc Lecuit. ✉email: marc.lecuit@pasteur.fr

nfectious disease symptoms can favour the transmission of pathogenic microorganisms and hence select genes that induce these symptoms (e.g. cough induced by *Mycobacterium tuberculosis*[1]). However, asymptomatic host colonisation can also favour microbial transmission, and thereby select for genes involved in host–microbe association that may also be involved in the development of opportunistic infections. *Listeria monocytogenes* (*Lm*) and *L. ivanovii* can cause infection in humans and other mammals including cattle[2,3], leading to fetal–placental infection, abortion and meningo-encephalitis, in contrast to other *Listeria* species which are non-pathogenic. *Lm* is known to alternate between a saprophytic and a host-associated lifestyle during which it expresses so-called virulence factors that mediate tissue invasion and within-host dissemination[4]. Most of these virulence factors are part of *Lm* core genome and therefore under purifying selection[5–7]. *Lm* most virulent clonal complexes are also the most adapted to mammalian gut colonisation[8] and *Lm* can be released from infected tissues back to the intestinal lumen[6,9,10], indicating that virulence may ultimately promote *Lm* faecal carriage and thereby play a major role in its dissemination.

*Lm* is a common contaminant of foodstuffs, and each human individual in Western countries is estimated to be exposed to *Lm* multiple times per year[11]. Yet the incidence of microbiologically proven invasive human listeriosis is extremely low, with 0.28 and 0.6 cases per 100,000 people in the United States and Europe, respectively[12,13]. This implies that in most cases, human exposure to *Lm* leads to either absence of clinically detectable infection and/or clinically silent gut colonisation, suggesting that *Lm* virulence genes are likely not selected for their capacity to induce clinically-overt disease. There have been reports of *Lm* asymptomatic faecal carriage among vertebrates including humans and cattle[14–23], and almost all studies have suggested that the prevalence of *Lm* carriage is below 1%[18,19,23]. However, these studies were mostly based on culture-based methods[18,19,23], which are less sensitive than molecular detection methods like PCR and sequencing[20,21,24]. Only one study suggested *Lm* faecal carriage to be above 1% by using PCR to detect *Lm*[20]. Large molecular studies on the distribution of *Listeria* species in mammals and the environment are not available[25–27].

In this work, we compare the distribution of pathogenic and non-pathogenic *Listeria* species in 16S rRNA gene datasets from diverse origins. We show that *Lm* is more host-associated than non-pathogenic *Listeria* species, and, in contrast to non-pathogenic *Listeria* species, is also present in the faeces of healthy humans, both in publicly available 16S rRNA datasets and in an independent cohort of asymptomatic individuals. A specific microbiota signature is associated with *Lm* faecal carriage, both in humans and experimentally inoculated mice, in which it precedes *Lm* faecal carriage. Asymptomatic carriage might represent an important outcome of *Lm* virulence, in addition to clinically-overt disease[28].

## Results

### *Listeria monocytogenes* is more host-associated than non-pathogenic *Listeria* species.
Ecological sampling is influenced by a priori assumptions about potential niches[29,30]. Here we circumvented this limitation by assessing *Listeria* species distribution in publicly available metagenomic datasets from the large MG-RAST database[31], to which high-quality metadata are associated, and retrieved 2490 full metagenomes and 11,907 16S rRNA high-quality datasets (see Methods). We assessed the impact of *Listeria* pathogenic potential on its ecological distribution by comparing the relative abundance (proportion of a species in a given sample, henceforth expressed as fractions, x-axis, Fig. 1) and prevalence (occurrence of a species in samples

of a given category, y-axis, Fig. 1) of the *Listeria* pathogenic species *Lm* and *L. ivanovii* to that of the non-pathogenic species *L. innocua*, *L. seeligeri* and *L. welshimeri*[32], termed together as *Listeria sensu stricto* (see Methods).

*Listeria sensu stricto* was detected in 26.06% 16S rRNA datasets (Figs. 1, 2a, b). Note that no positive result could be obtained using our approach (see Materials and Methods) analysing full metagenomes, in line with the relatively low abundance of *Listeria* species[33] and consistent with a higher sensitivity of 16S rRNA gene sequencing compared to full metagenome sequencing for a given sequencing depth[34]. *Lm* was most frequently present in soil (673/1700, ≥39.59%; mean relative abundance $1.2 \times 10^{-4}$), sludge (70/309, ≥22.65%), sediment (32/170, ≥18.82%) and host-associated samples (854/7695, ≥11.10%; mean relative abundance $9.0 \times 10^{-5}$). Only a few water samples (42/1980, ≥2.12%) and no air sample (0/53) were positive for any *Listeria* species (Figs. 1a, 2a for normalised data per category). Note that these *Lm* prevalence are likely an underestimate, as the sequenced 16S rRNA region varies among the samples studied and does not always allow to discriminate between *Listeria* species (see Method section), and that the distribution of undefined *Listeria* hits followed a similar distribution to those obtained from discriminant regions of the 16S rRNA gene (Fig. 2b). *Lm* was the most prevalent *Listeria* species in both soil and host-associated environments (Fig. 1a). Further, in samples where more than one *Listeria* species was present, *Lm* was significantly more abundant than other *Listeria* species, both in soil and hosts (Fig. 2c). We next investigated the *Listeria* species host range (Fig. 1b). *Lm* was found to be the most abundant (mean relative abundance $5.5 \times 10^{-3}$) and prevalent in cattle (80/1270; ≥6.30%), which have indeed been reported as a potential reservoir for *Lm*[26], especially hypervirulent clonal complexes[3,8,15]. We detected *Lm* in human samples at a similar prevalence to cattle (173/3338; ≥5.18%), but 40 times less abundantly (mean relative abundance $1.3 \times 10^{-4}$). *Lm* was also frequently found in chicken (mean relative abundance $3.6 \times 10^{-4}$, prevalence 28/552, ≥5.05%) and pig samples (mean relative abundance $4.7 \times 10^{-4}$, prevalence 48/300, ≥16%) but not that of goats (0/212), where only *L. ivanovii* was detected, consistent with the known association of *L. ivanovii* with small ruminants[35]. A high *Lm* prevalence in pigs and wild boars has been reported[36–38], and pigs might constitute an underappreciated niche for *Lm*. We next investigated the human sampling sites in which *Lm* was present. As expected for a foodborne pathogen, *Lm* was detected in faecal samples (108/2238, ≥4.83%; 108/1397, 7.73% in samples with discriminatory 16S rRNA sequences), but also in oral samples (7/108, ≥6.48%, Fig. 1c), consistent with reports that *Lm* may colonise both the gut and the oral cavity[39,40]. *Lm* was present in sputum (3/50, ≥6.00%) and skin samples (2/56, ≥3.57%) and absent in vaginal samples (0/30), but for these categories, only a few datasets were available for analysis. The non-pathogenic species *L. innocua* and *L. seeligeri* were not detected in any human-associated samples, while *L. ivanovii*, the only other pathogenic *Listeria* species, was detected, albeit far less than *Lm*, second most frequently in human stools (Fig. 1c).

### *Lm* faecal carriage is common in humans.
We aimed to replicate the result of frequent *Lm* carriage in humans independently and assessed *Lm* presence by *hly* PCR in the stools of a cohort of 900 healthy and a cohort of 125 diarrhoeic individuals (see Methods). It was detected in 10% (90/900) of healthy human stool samples and 20.8% (26/125) of diarrhoeic stools samples (Fig. 2d and Supplementary Tables 1 and 2). We confirmed that the sequence of the amplicon was that of *hly* in all of the samples we sequenced ($N = 10$) (Supplementary Data 1). The enrichment of *Lm* in

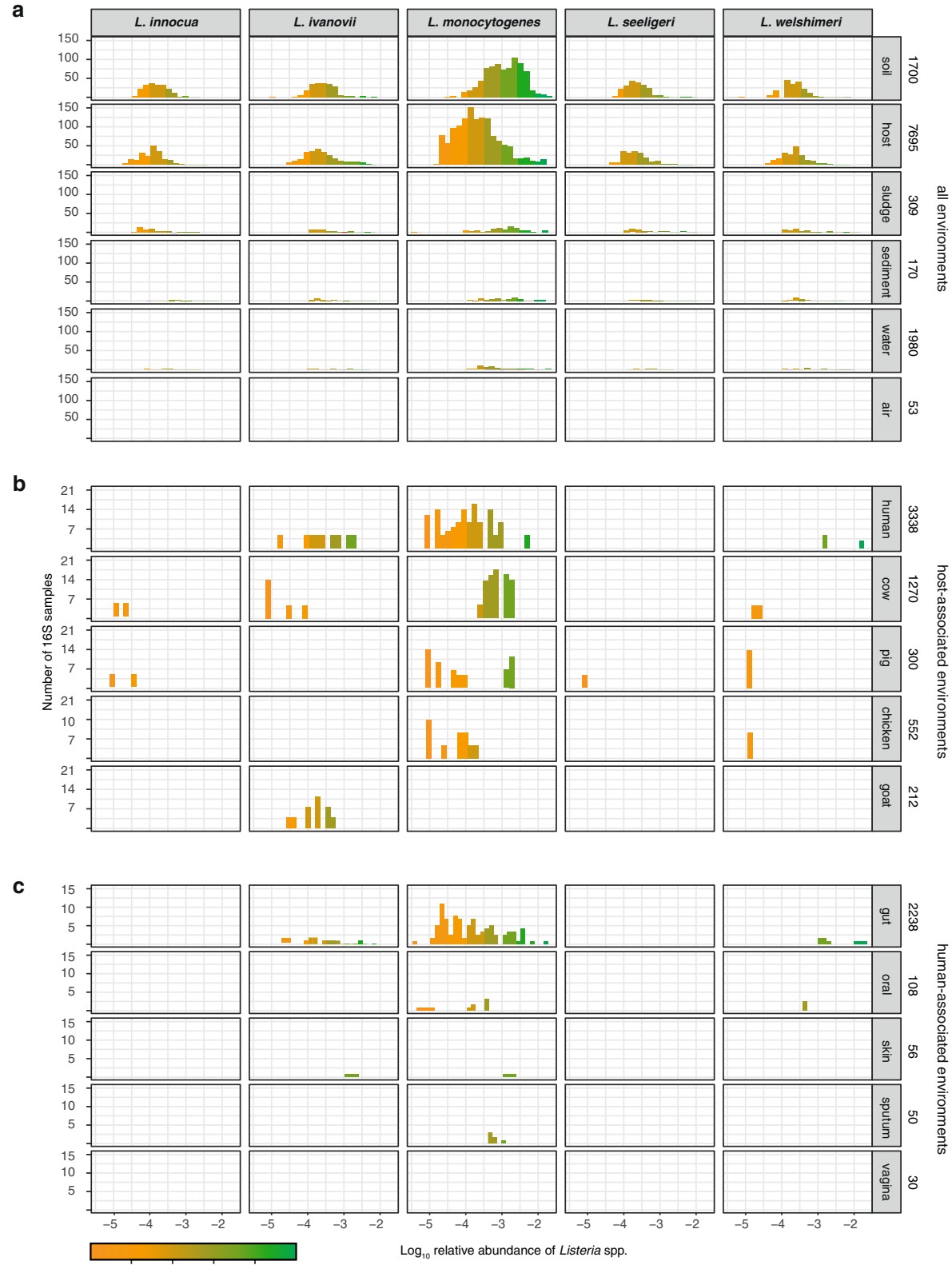

**Fig. 1 _Lm_ is more prevalent in host-associated environments than non-pathogenic _Listeria_ species.** Relative abundance and prevalence of _Listeria sensu stricto_ species in 16S rRNA gene datasets in **a** different environments, **b** in selected host datasets (i.e. farm animals and/or known _Lm_ reservoirs which were present >100x in our datasets) for which metadata detailing the host species were available and **c** from different sampling sites of healthy human hosts for which detailed metadata on body sampling site were available. Numbers on the right indicate samples per category.

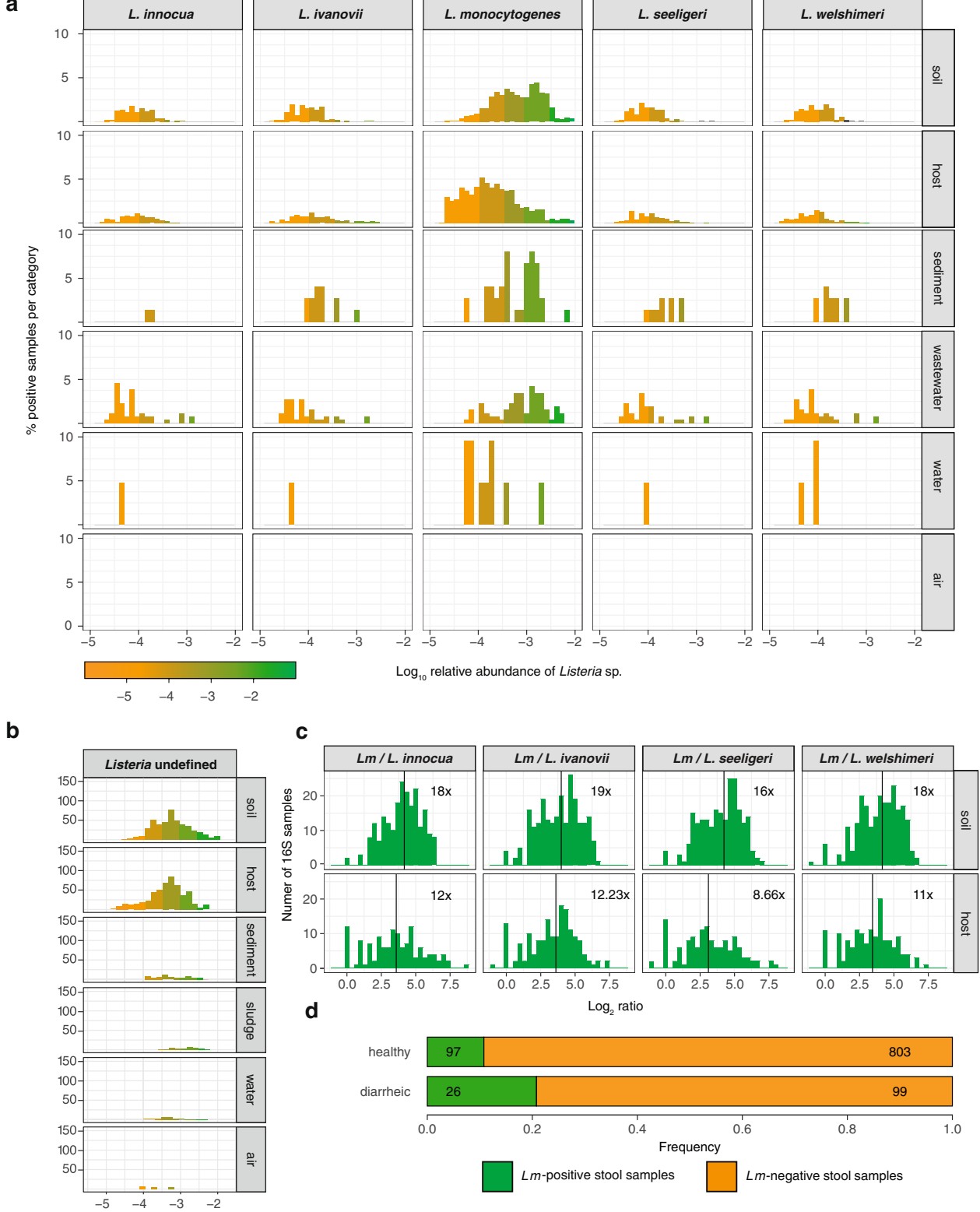

**Fig. 2 *Lm* is more abundant than other *Listeria* species in all environments and *Lm* carriage is common in healthy individuals. a** Same as Fig. 1a, normalised by category. **b** Relative abundance and prevalence of *Listeria* undefined in 16S rRNA gene datasets in different environments. **c** Log₂ of ratio of *Lm* to each other evaluated *Listeria* species in samples where the species co-occurred. Vertical lines and numbers indicate the mean of the distribution. **d** Prevalence of *Lm* in human faecal samples from healthy (*n* = 900) and diarrhoeic donors (*n* = 125) from France.

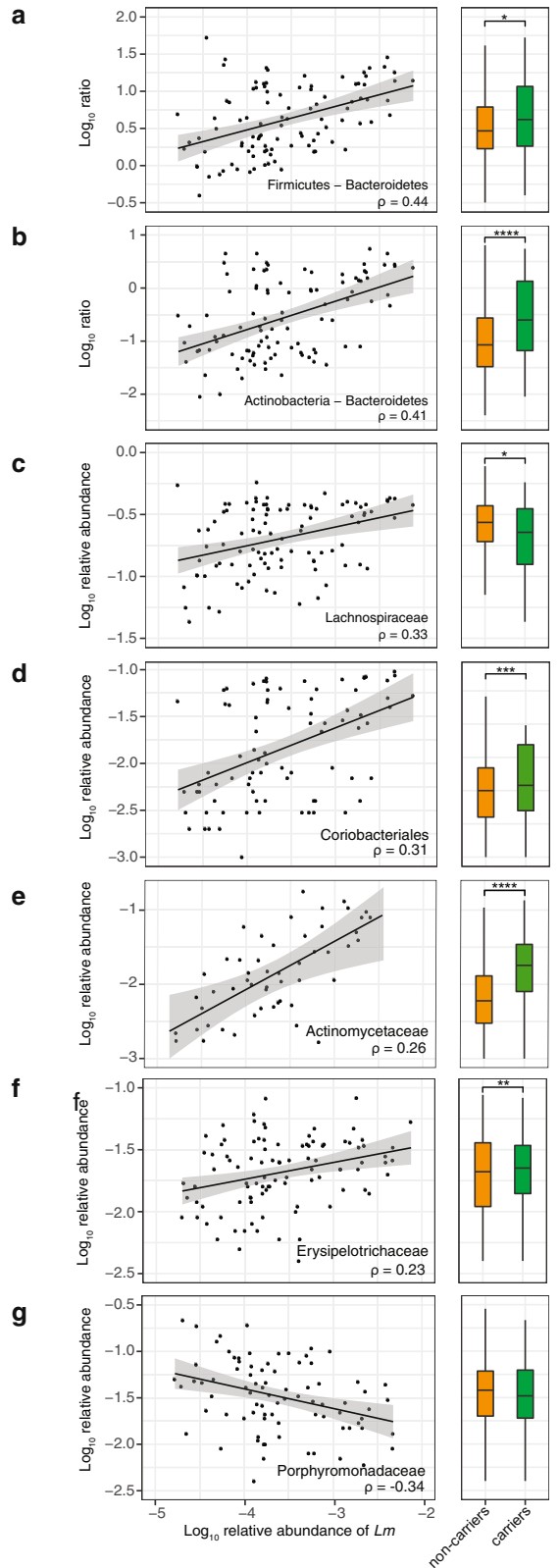

**Fig. 3 $Lm$ faecal carriage correlates with a specific microbiota signature in humans.** All significant correlations with more than 75 associated samples and rho >0.2 or < −0.2 between $Lm$ and commensal relative abundance in 108 healthy carrier (left panels) and comparison between carriers and non-carriers ($n_{non-carriers} = 2130$) for the same groups (right panels). The lines in the left panels corresponds to linear regression models and the grey area to their 95% confidence interval: **a** The ratio of Firmicutes to Bacteroidetes phyla (rho = 0.44, $p = 2.75 \times 10^{-5}$; non-carriers vs. carriers: 0.019; note that $Lm$ species was excluded when the relative abundance of Firmicutes was calculated), **b** The ratio of Actinobacteria to Bacteroidetes (rho = 0.414, $P = 6.1 \times 10^{-5}$; non-carriers vs. carriers: $P = 9 \times 10^{-15}$), **c** Lachnospiraceae (rho = 0.326, $P = 1.25 \times 10^{-3}$; non-carriers vs. carriers: $P = 0.026$), **d** Coriobacteriales (rho = 0.314, $P = 4.01 \times 10^{-2}$; non-carriers vs. carriers: $P = 0.000459$), **e** Actinomycetaceae (rho = 0.265, $P = 7.18 \times 10^{-11}$; non-carriers vs. carriers: $P = 3.093 \times 10^{-48}$), **f** Erysipelotrichaceae (rho = 0.226, $P = 4.51 \times 10^{-2}$; non-carriers vs. carriers: $P = 0.000361$), **g** Porphyromonadaceae (rho = −0.337, $P = 4.28 \times 10^{-3}$; non-carriers vs. carriers: $P = 0.215$). The rho values are Spearman correlation coefficients. Statistical comparison between carriers and non-carriers were performed with two-sided Wilcoxon rank-sum test with Benjamini–Hochberg correction for multiple test. * $P < 0.05$, ** $P < 0.01$, *** $P < 0.001$, **** $P < 0.0001$. For boxplots, the hinges represent the first and third quartile of the distribution. The whiskers extend from the hinge to the largest or smallest value no further than 1.5 x IQR from the respective hinge (where IQR is the inter-quartile range or distance between the first and third quartiles).

differential exposure to $Lm$-contaminated food[31]. Neither age nor gender was associated to an asymptomatic carriage (Supplementary Table 2). Of note, our assay cannot distinguish between dead and viable $Lm$ and the detection of $Lm$ DNA could therefore correspond to dead $Lm$.

***$Lm$ carriage correlates with a specific gut microbiota signature.*** The gut microbiota is a major line of defence against foodborne pathogens, and several commensals exert a protective effect against enteropathogens[43], including $Lm$[44]. $Lm$ also produces bacteriocins that can alter microbiota composition[45,46]. In order to assess if microbiota composition has an impact on $Lm$ faecal carriage in humans and vice versa, we investigated the relative abundance of microbiota taxonomic groups in MG-RAST human faecal samples. To take into account the compositional nature of data of different studies[47], we calculated the ratios between microbiota phylogenetic groups and $Lm$ abundance in the human microbiome datasets where $Lm$ is present (Fig. 3 and Supplementary Data 2 and 3). $Lm$ abundance correlated with the ratio of abundance of Firmicutes to Bacteroidetes phyla (Fig. 3a left), consistent with the observation that an increase of this ratio correlates with increased susceptibility to $Lm$[48]. This correlation is not due to $Lm$ itself, as this species was excluded when the relative abundance of Firmicutes was calculated. The ratio of Actinobacteria to Bacteroidetes also correlated with $Lm$ abundance (Fig. 3b left), and Actinobacteria were also significantly enriched compared to Firmicutes and Proteobacteria (Supplementary Data 2). $Lm$ abundance also correlated positively at the family and order levels with Lachnospiraceae (Fig. 3c left), Coriobacteriales (Fig. 3d left), Actinomycetaceae (Fig. 3e left), Erysipelotrichaceae (Fig. 3f left) and negatively with Porphyromonadaceae (Fig. 3g left). Erysipelotrichaceae have previously been reported to be elevated in asymptomatic *C. difficile* carriers, which suggests that loss of colonisation resistance is associated with this family[49]. In line with our results, a protective effect of Porphyromonadaceae has also been observed against *Salmonella enterica* serovar Typhimurium[50], *Enterococcus faecium*[51] and C.

diarrhoea samples ($\chi^2 = 11.702$, $P = 0.0018$, Benjamini–Hochberg correction) is consistent with the observation that $Lm$ can induce diarrhoea[41,42]. The difference in healthy asymptomatic donors in this cohort from France, relative to the 16S rRNA gene datasets from MG-RAST may be due to the different sensitivities of the two methods (targeted *hly* amplification versus total 16S amplification), and sample selection bias reflecting a potential

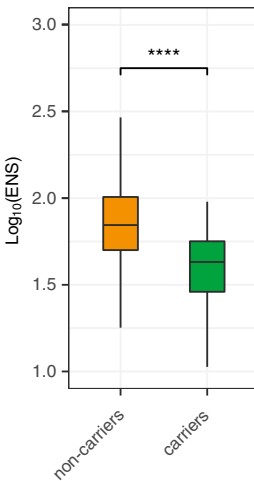

**Fig. 4 *Lm* faecal carriage correlates with low α-diversity in humans.** α-diversity, measured by ENS between carriers and non-carriers ($n_{carriers} = 108$, $n_{noncarriers} = 2130$, $P < 10^{-15}$). Statistical comparison was performed with a two-sided Wilcoxon rank-sum test. For boxplots, the hinges represent the first and third quartile of the distribution. The whiskers extend from the hinge to the largest or smallest value no further than 1.5 x IQR from the respective hinge (where IQR is the inter-quartile range or distance between the first and third quartiles). Points beyond this limit are shown.

difficile[52]. The aforementioned significant associations with *Lm* abundance in faecal carriers were also found significant between carriers and non-carriers (Fig. 3, right panels), with the exception of Porphyromonadaceae for which only a trend was observed (Fig. 3g right). For Lachnospiraceae, non-carriers showed a significantly higher prevalence than carriers (Fig. 3c right), reflecting that the comparisons between carriers and non-carriers are prone to study- and sample-dependent biases. Carriers also displayed less diverse microbiomes than non-carriers (Fig. 4), a finding consistent with the observation that α-diversity is also involved in colonisation resistance[43] against enteropathogens such as *C. difficile*[53], *Salmonella* or *Shigella*[54]. The overlap between the microbiota features associated with intestinal colonisation by *Lm* and other well-known gut-colonising bacteria is consistent with our finding that *Lm* is frequently present in stools of asymptomatic individuals.

**_Lm_ faecal carriage depends on gut microbiota.** *Lm* shedding from infected tissues back in the intestinal lumen may favour long-term faecal carriage[6,9,10], in line with the finding that the most virulent *Lm* clonal complexes are the most adapted to the mammalian gut[8], and the present observation that non-pathogenic species are not found in stool datasets retrieved from MG-RAST. To study *Lm* faecal carriage and its determinants experimentally, we inoculated mice intravenously with $5 \times 10^3$ CFUs of *Lm* belonging to the hypervirulent clonal complex-1[8,55]. We observed a cage-dependent faecal carriage in 3/7 cages (11/26 mice). *Lm* could be detected over 30-days post-inoculation, at a time when all mice were asymptomatic. We classified faecal carriage as either heavy (>$10^6$ CFU/g, six mice in two cages) or light (<$10^6$ CFU/g, four mice in one cage, together with one noncarrier mouse) (Fig. 5a). In four cages (15 mice), no *Lm* was detected in the faeces 30-days post-inoculation (Fig. 5a). All mice had fully recovered from symptoms developed in the three first days following inoculation as assessed by weight gain, independently of their carrier status (Fig. 5b) and no mice exhibited any detectable clinical sign thereafter, as assessed by daily observation after the inoculation. We also separated mice

and observed persistent faecal carriage, ruling out that it was resulting from coprophagy.

Co-housed animals tend to have similar microbiota[56], therefore the cage dependency of the observed differences in *Lm* carriage suggested that it was mediated by differences in gut microbiota composition. Indeed, heavy, light and noncarrier microbiota differed in microbial richness (α-diversity) and composition (β-diversity): heavy carriers' microbiota was less diverse than that of light and non-carriers (Fig. 6a), in line with results obtained in humans (Fig. 4). β-diversity analysis showed that faecal carriage groups differed also in composition (PERMANOVA $P < 0.001$): heavy carriers clustered separately from light and non-carriers (Fig. 6b). The difference between the light carrier group and the others reflected the higher homogeneity of the former (Fig. 6b), and the difference between non- and heavy carriers was mainly driven by a different composition in Bacteroidetes and Firmicutes: while OTUs classified as Porphyromonadaceae and Lachnospiraceae, respectively, were enriched in non-carriers, Bacteroidaceae were more present in heavy carriers (Figs. 6c, d and 7a–c).

We next investigated whether these differences in microbiota precede or result from *Lm* carriage (Fig. 8a, b). To this end, we compared microbiota 16S rRNA gene composition before *Lm* inoculation and 30-days post-inoculation. *Lm* inoculation only marginally affected microbiota composition, with only few OTUs (16/710) changes upon *Lm* inoculation. In sharp contrast, very significant OTUs differences were observed between carriers and non-carriers (685/1660) (Fig. 8b). The observation that the β-diversity difference observed between heavy and noncarrier microbiota preceded *Lm* inoculation (Axis 1 in Fig. 8a) suggests that it plays a causative role in the establishment of *Lm* carriage. To demonstrate that *Lm* carriage is indeed driven by the microbiota composition, we aimed to perturb it and assess if it would affect their permissiveness to *Lm* faecal carriage. As we had shown that a lower microbiota α-diversity is associated with *Lm* faecal carriage both in mice and human 16S rRNA gene datasets, we treated mice with a broad range antibiotic oral cocktail shown to lower α-diversity[57] and let the microbiota regrow for 4 weeks. We then inoculated these mice and PBS-treated littermates with *Lm*. α-diversity was indeed significantly lower in antibiotic-treated mice at the time of inoculation, compared to PBS-treated mice (Fig. 8c). Both antibiotic- and PBS-treated mice lost weight upon *Lm* inoculation and recovered similarly (Fig. 8d). However, while none of the PBS-treated mice carried *Lm* in their faeces at 30-days post-inoculation, 6/7 antibiotic-treated mice carried *Lm* (Fig. 8e, $P = 0.0013$). These results demonstrate that permissiveness to *Lm* carriage is driven by the composition of the gut microbiota.

## Discussion

Here we have shown that *Listeria* faecal carriage correlates with virulence: it is common in pathogenic *Listeria* species while it is rare in non-pathogenic species. This finding suggests that establishing faecal carriage, potentially through clinically silent tissue invasion and reseeding of the gut lumen via the gallbladder[9], is a potential function of virulence genes in *Lm* ecology. Indeed, clinically-overt *Lm* infection is actually rare and is not involved in inter-human horizontal transmission[58]. This also implies that humans are not a focal host[59] for *Lm*. Consistent with this, *Lm* is more prevalent and abundant in cattle than in human stools, which is also in line with our recent report that hypervirulent *Lm* clonal complexes are associated to cattle and dairy products[8]. We also now report that the phylogeography of the hypervirulent *Lm* clonal complex-1 is linked to cattle global trade and farming[60]. Taken together, these observations strongly suggest

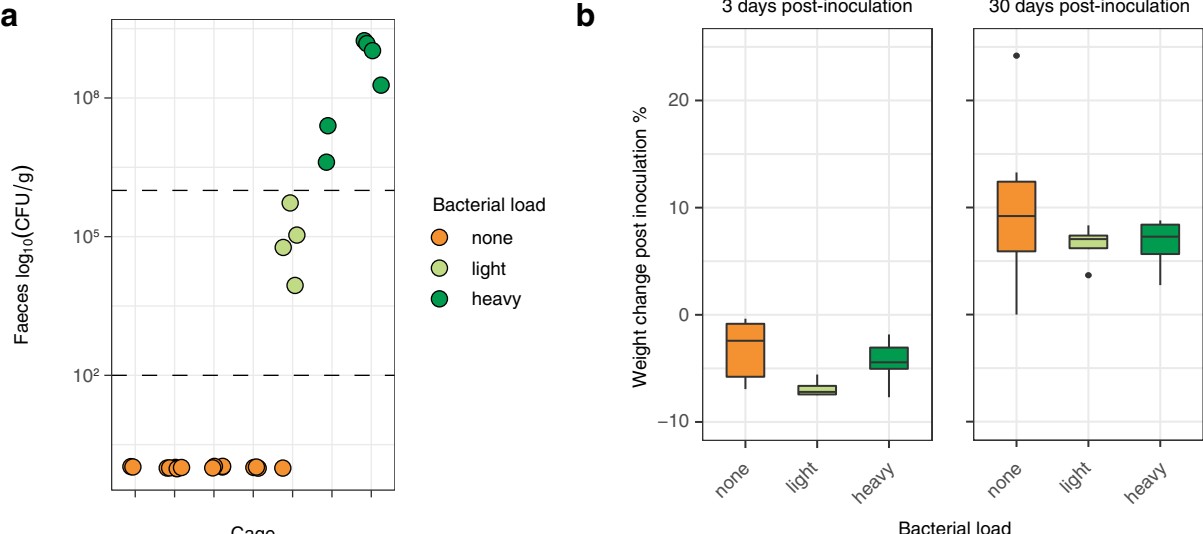

**Fig. 5 *Lm* faecal carriage in mice is cage-dependent. a** CFU/g of the stool of male mice 30 days after an *iv* challenge with *Lm* at $5 \times 10^3$ CFU from different cages (2–6 mice per cage). Colour indicates carriage group (<100 CFU/g: none, 100–$10^6$ CFU/g: light, >$10^6$ CFU/g: heavy). Horizontal lines indicate the threshold between the groups. **b** Body weight change of mice after inoculation at 3 days post-inoculation and 30 days post-inoculation according to their carriage group ($n_{none} = 16$, $n_{low} = 4$, $n_{high} = 6$; 3 dpi: none vs. heavy $P = 0.37$, low vs. heavy $P = 0.11$; 30 dpi: none vs. heavy $P = 0.37$, low vs. heavy $P = 0.91$). The statistical comparison was performed with a two-sided Wilcoxon rank-sum test. For boxplots, the hinges represent the first and third quartile of the distribution. The whiskers extend from the hinge to the largest or smallest value no further than 1.5 x IQR from the respective hinge (where IQR is the inter-quartile range or distance between the first and third quartiles). Points beyond this limit are shown.

that cattle constitute a major reservoir where *Lm* virulence is selected for.

We also found that *Lm* is the predominant *Listeria* species in the environment, where it is a saprophyte. *Lm* persistence in food processing plants, away from its vertebrate hosts, is associated with loss of virulence[7,8,55]. That *Lm* is found more abundantly in soil, sludge and sediments than non-pathogenic species (Fig. 1a) suggest that *Lm* regularly transits between its hosts via these environments, while maintaining its host-association capacity, which is mediated by its virulence genes. *Listeria* host-association capacity, therefore, appears as a trait that ensures the ecological success of *Lm* and *L. ivanovii* relative to other *Listeria* species. This does not exclude that *bona fide* virulence factors also contribute to *Lm* saprophytic lifestyle, as shown for ActA, which is involved in biofilm formation[61]. The relative lower prevalence in the environment of non-virulent *Listeria* species *L. innocua*, *L. seeligeri* and *L. welshimeri* which derive from the common virulent ancestor of *Lm* and *L. ivanovii*[62] suggest that they either (i) successfully colonise an environment not sampled in this study, and/or (ii) lost their focal host, and/or (iii) lost their host-association capacity, similar to *Lm* isolates associated with food processing plants which have lost or are in the process of losing virulence[7,8,55]. Future research will have to study the contribution of virulence and host-association to the overall ecological success of *Lm* and other microbial species which, as *Lm*, are widespread in the environment. It will also have to address the relative contribution of host and *Lm* genetics, food habits and intestinal microbiota to the asymptomatic faecal carriage of *Lm*.

## Methods

**Screening of Listeria sp. in 16S rRNA datasets**. A summary of the study workflow is represented in Supplementary Fig. 1. We collected 13,749 16S rRNA amplification datasets from MG-RAST from studies with >5 and <50 samples, for studies containing host samples <250 (last accessed: November 2017) as described in ref. [63]. When more samples were available, we randomly selected 50 or 250 samples, respectively. We removed those containing non-ribosomal data or less than 2000 sequences using SSU-align v.1.01[64]. This left us with a total of 11,907 rRNA datasets (Supplementary Data 4, 5 for a list of all host species analysed in this study). Sequences shorter than 60 bp were removed. 16S rRNA sequence

datasets were re-aligned using mafft v. 7.407[65], and trimmed using trimal v.1.4[66] using the 'automated1' algorithm. The resulting trimmed sequences were then clustered within each sample at 99% identity and 90% coverage using the uclust algorithm from usearch v. 10.0.240[67]. Non-redundant reads which were present >3 were considered in the analysis. A representative sequence of each cluster was defined according to the distance to the cluster centroid. Henceforth, we will call these our environmental dataset.

To identify *Listeria* ssp. in the environmental dataset we used a maximum likelihood approach. First, *Listeria sensu stricto* 16- S rRNA reference sequences (accession numbers X56153, X98527, DQ065846, DQ065845 and X98528) were obtained from Genbank[68] and aligned using mafft with the 'linsi' algorithm. The resulting multiple sequence alignment was trimmed using trimal v.1.4. Phylogenetic reconstruction was then performed using IQ-tree v.1.6.5[69] using the GTR model (according to the model test) and 1000 rapid bootstrap iterations. The resulting tree was manually pruned to leave only one representative member of each clade. Environmental sequences were then classified as potential *Listeria* candidates by mapping them against the multiple sequence alignment using the '-addfragments' algorithm of mafft. Sequences with at least 90% identity and 90% coverage to one reference member were kept for further analyses, or otherwise were discarded. The remaining sequences were then assigned to one of the branches of the phylogenetic tree using the evolutionary placement algorithm implemented in RAxML v. 8.2[70]. Environmental sequences assigned to any terminal branches with a maximum likelihood of 0.6 or higher, were classified as the specific *Listeria* species. Otherwise, they were classified as '*Listeria* undefined'. Note that this was the case for sequences with a non-discriminative amplicon region at the species level, e.g. V3-V4. In this work, we focused on all *Listeria sensu stricto* species, which are frequently found in the environment (*Lm*, *L. ivanovii*, *L. innocua*, *L. seeligeri* and *L. welshimeri*). We did not include the closest non-pathogenic relative of *Lm*, *L. marthii* since it is only rarely sampled in any environment[71].

The remaining non-*Listeria* representative sequences were used to construct a global catalogue of operational taxonomic units (OTUs). To do so, the representative sequences of all datasets were grouped and clustered together at 97% identity using usearch, and the frequency of each OTU was calculated on each dataset. Finally, OTU representatives were taxonomically classified at the genus level using the RDP classifier[72]. At the same time, we defined the α-diversity of each dataset as the Expected Number of Species (ENS). To do so, we did calculate the Shannon diversity index (H′) (1):

$$H' = -\sum_{i=1}^{R} p_i \ln(p_i)$$

where $p_i$ is the relative frequency of a specific species in the dataset (the number of sequences associated with the species divided by the total number of sequences assigned to species), and R is the number of datasets. We calculated the ENS as the

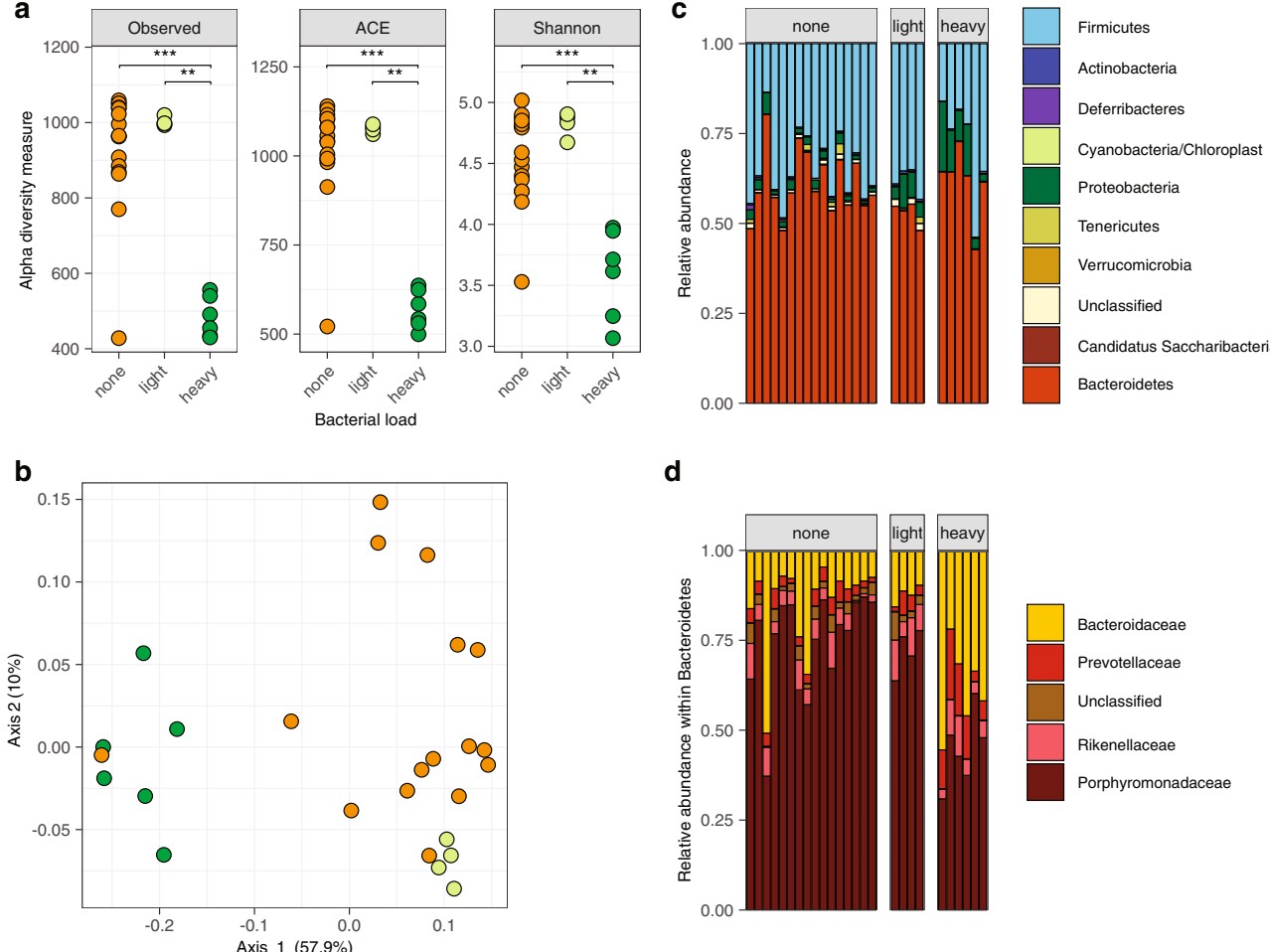

**Fig. 6 Lm faecal carriage correlates with a specific microbiota signature in mice. a** Carriage groups differ in α-diversity, measured by observed species (left), abundance-based coverage estimate (middle) and Shannon index (right), (Observed: none vs. heavy: $P = 0.00017$, light vs. heavy: $P = 0.014$, ACE: none vs. heavy: $P = 0.00026$, light vs. heavy: $P = 0.0095$, Shannon: none vs. heavy: $P = 0.0001$, light vs. heavy: $P = 0.0095$). **b** β-diversity of mice microbiomes using MDS and Bray–Curtis distance. The colour indicates the carriage group (<100 CFU/g: none, 100–$10^6$ CFU/g: light, >$10^6$ CFU/g: heavy). All groups differed in composition (PERMANOVA overall $P = 0.001$, heavy/none $P = 0.006$, heavy/light $P = 0.0075$, light/none $P = 0.031$, with Benjamini–Hochberg correction). Light carriers were more homogeneous than other groups (permutation test for homogeneity of multivariate dispersion, heavy/none, $P = 0.246$, heavy/light $P = 0.0160$, light/none $P = 0.0193$, with Benjamini–Hochberg correction) **c** Microbiota composition of mice from Fig. 2c at phyla level and **d** family level within the Bacteroidetes phylum. Statistical comparison performed with two-sided Wilcoxon rank-sum test. *$P < 0.05$, **$P < 0.01$, ***$P < 0.001$, ****$P < 0.0001$.

exponential of the Shannon diversity (2):

$$\mathrm{ENS} = e^{H'}$$

**Statistical analyses on the 16S rRNA datasets**. Given the lack of quantitative data associated with the public datasets, data were treated as compositional. In order to avoid methodological biases when comparing the datasets, only comparisons between ratios were performed. To assess associations between $Lm$ abundance and other taxonomic groups, and given the non-parametric nature of the microbiome data, both Spearman correlation and Wilcoxon rank test were used to compare between the relative abundance of $Lm$ and the different taxonomic ratios. Benjamini–Hochberg correction was used to avoid type I statistical error.

**Screening of Listeria sp. in full metagenomes**. We followed the approach described in Garcia-Garcera et al.[73]. In brief, we retrieved around 3000 metagenomes (Supplementary Data 6) from MG-RAST. To characterise the presence of $Listeria$, reads were mapped against a discriminative concatenate of genes, here the seven housekeeping genes used for multilocus sequence typing in $Lm$[74]. Since no reads could be mapped to these genes concatenate, no further downstream analysis has been performed. The absence of $Listeria$ in full metagenomes can either be due to the low relative abundance of $Listeria$ per se which does not permit it to be identified, or previous filtering of reads from species with low abundance.

**Detection of Lm in human faecal samples**. Determination of faecal carriage of $Lm$ was performed by PCR amplifying $hly$[24]. We evaluated the performance of this method using artificial samples that mimic natural stool (Supplementary Table 1). Briefly, tenfold dilutions of the ATCC $Lm$ strain (ATCC BAA751) in saline buffer ($10^8$ to $10^1$ CFU/mL) were diluted in a 1:1 ratio in a PCR-negative stool sample conserved on eNat (Copan, Italy) before extraction. Extraction was performed with EasyMag (bioMérieux, Marcy-l'Etoile, France) according to manufacturer's recommendations. PCR assays were performed in triplicate on a CFX96 system (BioRad, CA, USA) as described[24]. $Lm$ was considered present when at least two of three PCR assays were positive. $Lm$ detection threshold was $10^6$ CFU/ml of stool.

Tested stool samples originated from two cohorts collected and stored on eNat (Copan): (i) the Hepystool cohort that included samples ($n = 900$ samples, 2015–2016) from non-diarrhoeic patients (inclusion criteria described in ref. [75]) and (ii) stool samples from diarrhoeic patients, received at the Infectious Agents Department of the University Hospital of Poitiers, France. DNA was extracted on EasyMag (bioMérieux, Marcy-l'Etoile, France) according to manufacturer's recommendations then amplified in triplicate. All samples which were at least once positive on the first triplicate were subjected to a second triplicate and were considered as positive when again detected at least once. The significance of the associations of the items listed in Supplementary Table 2 with $Lm$ presence was assessed as follows: Student's $t$-test for quantitative data and $\chi^2$-test for qualitative data except for birthplaces where a Fisher exact test was used.

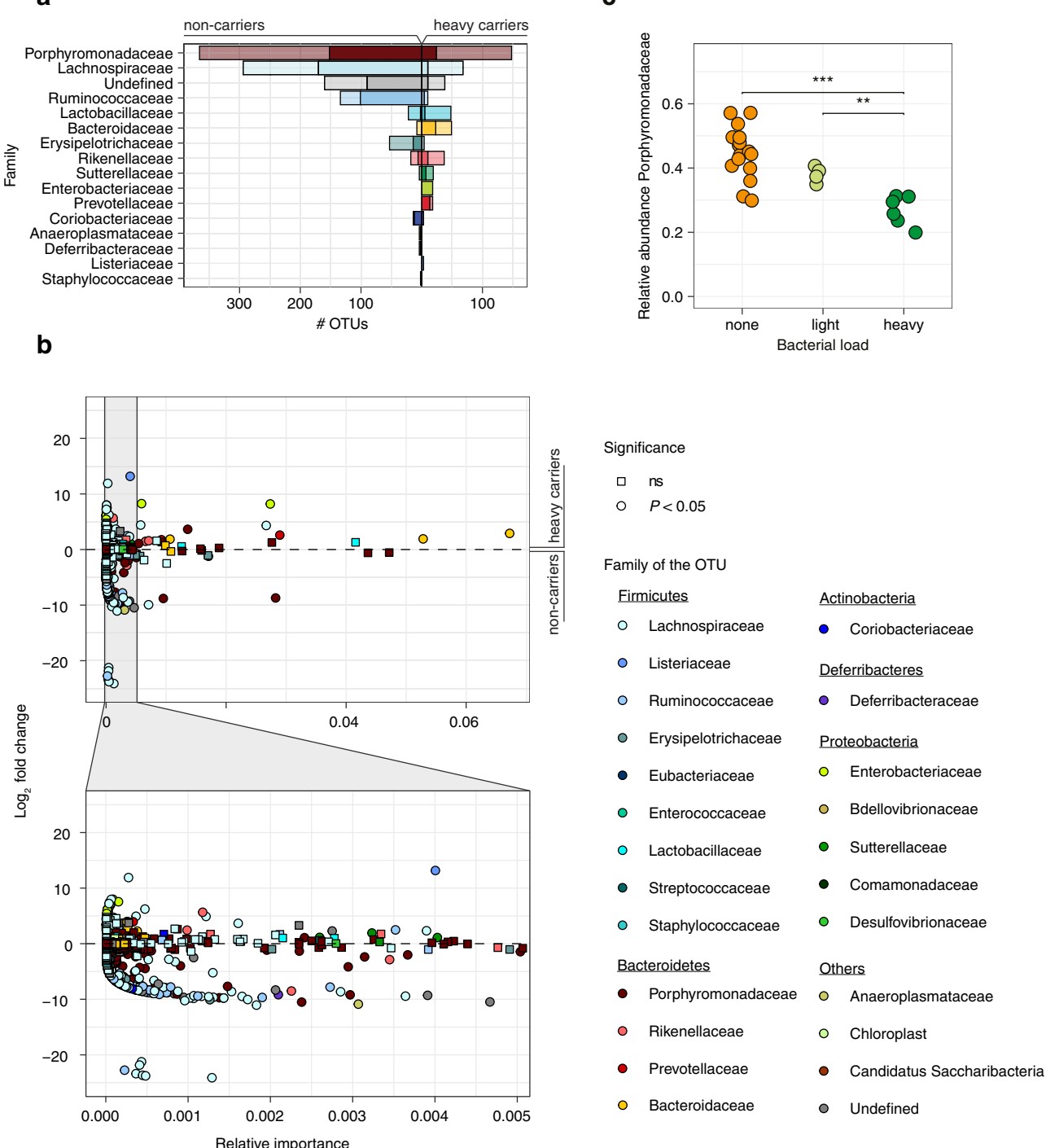

**Fig. 7 Microbiota composition in mice differs according to *Lm* faecal carriage. a** Count of all OTUs enriched in non-carriers (left) or heavy carriers (right) identified by DESeq2. Opaque bars represent significantly different OTUs (after Benjamini–Hochberg correction). **b** Mean abundance and fold change of all OTUs between carrier and noncarrier mice determined by DESeq2. Opaque points significantly differ between the two groups (*P* after Benjamini–Hochberg correction). **c** Relative abundance of Porphyromonadaceae in 16S rRNA data from mice from different carriage groups ($n_{none} = 16$, $n_{low} = 4$, $n_{heavy} = 6$, none vs. heavy: $P = 0.0001$, light vs. heavy: $P = 0.0095$). The statistical comparison was performed with a two-sided Wilcoxon rank-sum test. \*\**P* < 0.01, \*\*\**P* < 0.001.

Note that no culture-based identification was applicable given that the eNat protocol conserves nucleic acid and is bactericidal within 30 min (see manufacturer's instructions for details). We sequenced the *hly* gene from position 327 to 829 with the primers CAAAATAATGCAGACATCCAAG and CTTTAGTAACAGCTTTGCCG for a subset (*N* = 10) of *Lm* positive samples to ensure the specificity of the *hly* PCR assay. Sequences from Sanger

sequencing with CAAAATAATGCAGACATCCAAG and EGD-e *hly* sequence retrieved from https://bigsdb.pasteur.fr/listeria/listeria.html were aligned with clustal omega[76].

**Mouse colonisation experiments**. Seven- to 11-week-old male mice (C57BL/6 mEcad E16P KI[77]) were infected intravenously in the tail vein as previously

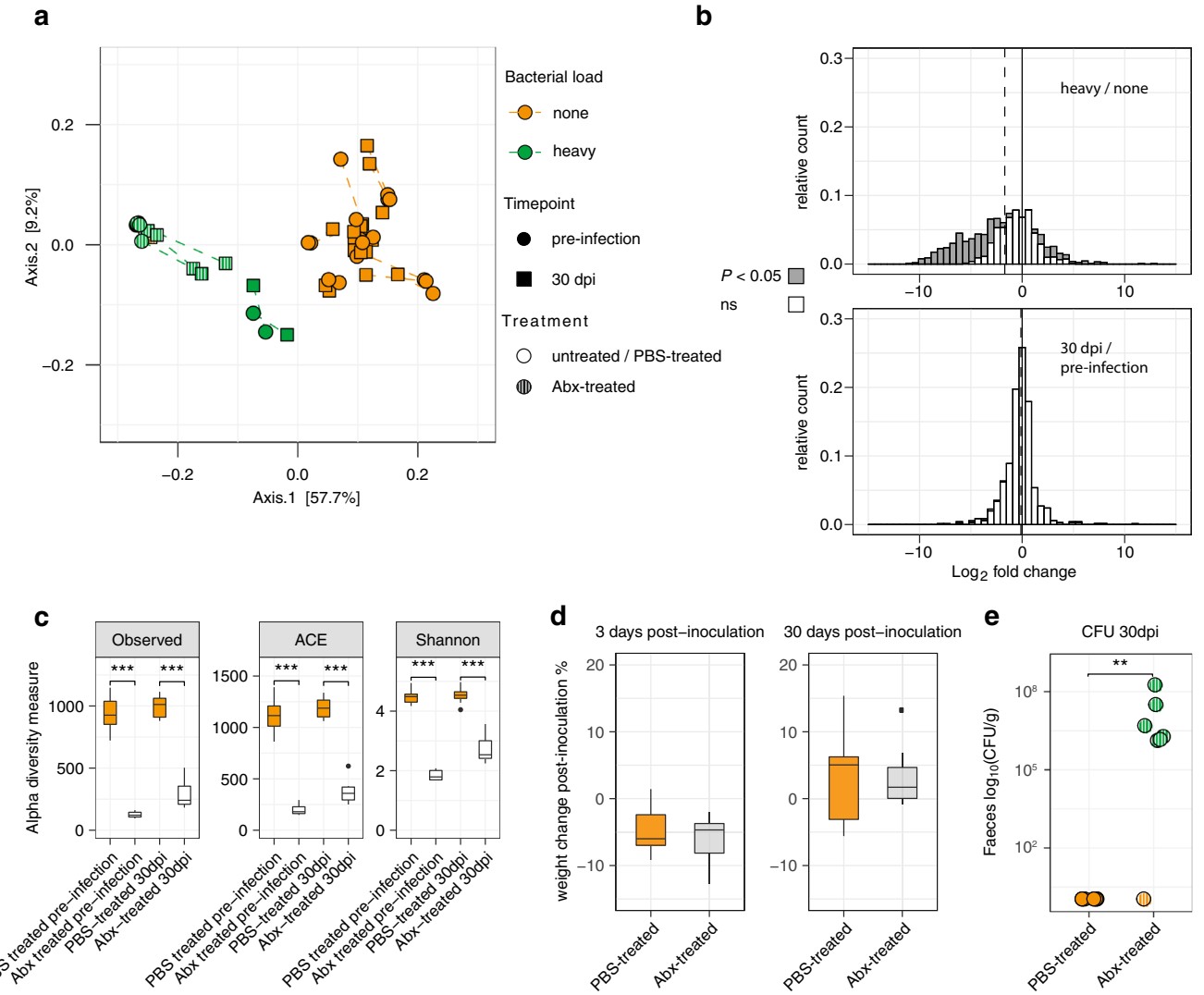

**Fig. 8 *Lm* faecal carriage depends on the microbiota. a** Comparison of microbiota pre-inoculation and 30-days post-inoculation. β-diversity of mice microbiomes using MDS and Bray–Curtis distance. The colour indicates the carriage group (<100 CFU/g: none, >10⁶ CFU/g: heavy), the shape of the timepoint (round: Pre-inoculation, square: 30-days post-inoculation) and opacity of the treatment (plain: untreated or PBS-treated, striped: antibiotic treated) **b** Distribution of fold change determined by DESeq2 of OTUs between heavy and non-carriers (top) and pre- and post-infection (bottom). Dark bars indicate significantly differently present OTUs identified by DESeq2 ($P < 0.05$ after Benjamini–Hochberg correction). The dashed bar indicates the median of the respective distribution. **c** Comparison of α-diversity, measured by observed species (left), abundance-based coverage estimates (middle) and Shannon index (right) between antibiotic- and PBS-treated mice 4 weeks after antibiotic treatment/before inoculation and 4 weeks after *Lm* inoculation. ($n_{Abx-treated} = 7$, $n_{PBS-treated} = 9$, PBS-treated vs. Abx-treated pre-infection $P = 0.00017$, 30dpi $P = 0.00017$). **d** Body weight change of mice at 3 days post-inoculation and 30-days post-inoculation according to their treatment group ($n_{PBS-treated} = 9$, $n_{Abx-treated} = 7$, PBS-treated vs. Abx-treated 3 dpi: $P = 0.76$, 30dpi: $P = 0.84$). **e** CFU/g of the stool of female mice 30 days after an *iv* challenge with *Lm* at $5 \times 10^3$ CFU. Mice were either treated with antibiotics or PBS 4 weeks prior to inoculation. Statistical comparison performed with two-sided Wilcoxon rank-sum test (PBS-treated vs. Abx-treated: $P = 0.0013$). The statistical comparison was performed with a two-sided Wilcoxon rank-sum test. For boxplots, the hinges represent the first and third quartile of the distribution. The whiskers extend from the hinge to the largest or smallest value no further than 1.5 x IQR from the respective hinge (where IQR is the inter-quartile range or distance between the first and third quartiles). Points beyond this limit are shown.

described[77]. Fraternities were kept together in a cage during the whole experiment, except when separated to exclude that carriage was due to coprophagy. To quantify carriage at 30-days post-inoculation, faeces were collected from each individual mouse and weighted before being homogenised in 2 ml of PBS. CFU count was performed by serial dilution of homogenised faeces on ALOA plates (Bio-mérieux) as described in ref. [8]. Separate faecal pellets were collected pre-inoculation and/or 30-days post-inoculation and stored at −20 °C for DNA extraction for 16S rRNA gene sequencing. DNA was extracted with DNeasy PowerSoil Pro Kit (Qiagen).

For experiments with antibiotic treatment, 5-week-old female mice (C57BL/6 mEcad E16P KI[77]) received orally every 12 h during 4 days PBS (as control) or the following antibiotics (similarly to[57]): ampicillin (100 mg/kg, Sigma-Aldrich A9618), vancomycin (50 mg/kg, Sigma-Aldrich 75423), metronidazole (100 mg/kg,

Abcam ab141218), neomycin (100 mg/kg, Gibco 11811-031), amphotericin B (1 mg/kg, ApexBio Technology B1885). The mice were inoculated, as described above, 4 weeks after the antibiotic treatment.

**16S rRNA gene analysis in mice**. DNA from faeces has been isolated with DNeasy PowerSoil Kit (Qiagen, Cat. No. 47016) accordingly to the manufacturer's instructions. The V4 region has been amplified and sequenced with the primers CCTACGGGNGGCWGCAG and GACTACNVGGGTWTCTAATCC using the Illumina MiSeq workflow at the biomics platform at the Institut Pasteur, Paris. Analysis have been performed with micca v.1.7.2[78], using the RDP classifier[79] and unoise3 for clustering[80]. Forward and reverse reads were merged with a minimum overlap of 100 bp and 30 maximum allowed mismatches. Forward and reverse primers were removed and reads were trimmed to 400 nucleotides

using the micca workflow. Reads with an expected error rate above 0.75% were excluded. Reads were grouped in sequence variants by unoise3[80] and chimeric sequences were removed. Sequence variants were classified with RDP[79], which uses VSEARCH to match sequences with the reference database[81]. Statistical were performed with R and the phyloseq v.1.34.0, vegan and microbiome libraries[82,83]. α-diversity has been calculated by the number of observed species, abundance-based coverage estimator (ACE)[84] and Shannon index[85]. β-diversity between samples has been calculated with MDS of Bray–Curtis dissimilarities[86]. PERMANOVA and homogeneity between microbiome groups were calculated with Adonis and Betadispers from the vegan library[87]. Differentially present OTUs between groups were identified with DESeq2 v.1.30.1[88]. To identify OTUs that differ between pre-inoculation and 30 dpi, the '~mouse_id + timepoint' design was used.

**Housing conditions of mice**. Animals were maintained in a facility which is licensed by the French Ministry of Agriculture (agreement B 75-15-01, 04, 05, 06, 07, 08, 09, 11, A 75-15-13 dated 22 May 2008 and A 75-15-27 dated 12 November 2004). The facility has central air conditioning equipment which maintains a constant temperature of 22 ± 2 °C. Air is renewed at least 20 times per hour in animal rooms. Light is provided with a 14:10 h light:dark cycle (6:30 a.m. to 8:30 p.m.). Animals were kept in polypropylene or polycarbonate cages which comply with European regulations in terms of floor surface per animal. All cages are covered with stainless steel grids and non-woven filter caps.

**Reporting Summary**. Further information on research design is available in the Nature Research Reporting Summary linked to this article.

## Data availability

Primary sequencing data are available on the Sequence Read Archive under the entry PRJNA642013. 16S rRNA datasets for screening of *Listeria* sp. were retrieved from MG-RAST.

## Code availability

Code for screening of a microbe of interest in 16S rRNA gene data is published in ref. [73].

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

## Acknowledgements

We thank Hélène Bracq-Dieye for technical help, Georges Michel Haustant, Cédric Fund, Elodie Turc, Laure Lemée, Biomics Platform, C2RT, Institut Pasteur, Paris, France, supported by France Génomique (ANR-10-INBS-09-09) and IBISA for 16S sequencing, Auguste Fourneau and Amandine Brunet for PCR assays and Sandrine Isaac, Henrik Salje and Olivier Disson for critical reading. L.H. is supported by the Pasteur-Paris University (PPU) International Ph.D. Programme, funded by the European Union's Horizon 2020 research and innovation programme under the Marie Sklodowska-Curie grant agreement No 665807, the 'Ecole Doctorale FIRE-Programme Bettencourt' of the CRI Paris and the Fondation Pasteur Suisse. M.L. laboratory is funded by Institut Pasteur, Inserm, the European Research Council and Laboratoire d'Excellence Integrative Biology of Emerging Infectious Diseases. M.L. is a member of Institut Universitaire de France.

## Author contributions

M.L. initiated and coordinated the project. L.H., A.M., M.G.-G. and M.L. designed the study. M.G.-G. collected and analyzed the public metagenomic data. C.B. and M.P. assessed the prevalence of *Lm* in stool donors' cohorts. L.H. conducted the in vivo experiments in mice, together with S.H.A.N. L.H. conducted the mouse 16S rRNA analysis. L.H. and M.L. wrote the manuscript, M.G.-G. and A.M. commented and edited on it.

## Competing interests

The authors declare no competing interests.

## Ethical statement

Animal experiments were performed according to the Institut Pasteur guidelines for laboratory animals' husbandry and in compliance with European regulation 2010/63 EU. All procedures were approved by the Animal Ethics Committee of Institut Pasteur, authorised by the French Ministry of Research and registered under #14644-2018041116183944. The stool donor cohorts received ethical approval from the regional Committee for the Protection of People (CPP Ouest III) and from the National Commission for Protection of Personal data on 23 November 2015. All patients were informed before inclusion and their consent was obtained before analysis.
