## [Peer Review File · Nature Communications]

REVIEWER COMMENTS

Reviewer #1 (Remarks to the Author):

This paper presents the results of 3 of complementary studies addressing the relationship between prevalence of *Listeria monocytogenes* (LM) and the microbiota in (i) publicly available, previously sequenced environmental samples, (ii) among a cohort of 900 healthy and diarrheal human stool samples and (iii) an experimental mouse model. The paper is well-written, and the experiments and analysis of the data obtained human LM carriage and the experimental mouse model seem appropriate and do seem to support that (potentially) asymptomatic carriage of LM may be the factor driving purifying selection on virulence genes in this (mostly environmental) pathogen.

The authors have addressed my concerns/comments of the first version of this manuscript adequately, and I think the current revised manuscript is a great improvement, mainly in clarity of the methodology and thus reproducibility and transparency of the (bioinformatics) analyses.

Henk C. den Bakker

Reviewer #2 (Remarks to the Author):

The manuscript has addressed several of the methodological issues brought up by the previous reviewers; however, concerns remain regarding certain methods and their potential outcomes, as well as regarding the interpretation of the findings.

(1) The manuscript's key conclusion that "asymptomatic faecal carriage, rather than disease, exerts purifying selection on Lm "virulence genes" remains unsupported. Asymptomatic fecal carriage may

indeed exert purifying selection, but this cannot be concluded from the current data, neither can it be concluded that it is the driving force. And where, in this manuscript, is the genetic analysis of virulence genes during and following fecal carriage, to support purifying selection on these genes? Previous work with oral infection models (e.g., Roll and Czuprynski 1990, and Ref 6 by Louie et al) and mutants in virulence genes showed that upon oral inoculation LM could cause systemic infection only if it harbored intact virulence genes such as hly and actA. Ref 6 has shown that only LM with intact hly and actA could colonize the gallbladder (subsequent to liver colonization), from there seeding the intestine. But these were actual systemic infections, not asymptomatic carriage. This means, that the purifying selection was still exerted at the level of disease (i.e., ability to colonize systemically, then colonize and grow in the gallbladder). In the current murine model which involved IV infection of the mice, LM likely entered the intestine following spillover from the liver to the gallbladder, i. e., following true systemic infection. Hence in this model as well, purifying selection is acting at the infection-disease level rather than at the level of fecal carriage (which is here taken as proxy for intestinal colonization, though this was not validated).

(2) In the above context, even though “Asymptomatic” faecal carriage is being repeatedly mentioned, L171-6 also mentioned that “all mice recovered from infection as assessed by weight gain”; this suggests the presence of symptoms. Similarly, L208-9 indicates “Both antibiotic- and PBS-treated mice lost weight upon Lm inoculation”. The animal model system here does not support “asymptomatic carriage” because symptoms (e.g., weight loss), were observed.

(3) L170-8, were any assessments done on LM in liver, spleen or gallbladder in the mice? Also, any other symptoms besides weigh loss? Authors should also elaborate as to whether the weight loss was accompanied with fever and lack of appetite.

(4) Authors, consider clarifying why an IV model was employed to assess intestinal colonization, instead of available oral inoculation models such as mentioned above. An oral model would be clearly advantageous for investigating the host and bacterial outcomes of ingestion of LM via foods. It would require suppression of the host interference with the gut microbiota, e.g., via antibiotic treatment, but antibiotic treatments were also used in the current model to make the gut more permissive to LM.

(5) The PCR-based defection of LM in the human stool samples remains problematic. Even though in their Response the authors indicate that the PCR prevalence was similar in the current study and Grif et al., the data in the manuscript indicate 10% in healthy humans and even twice that in diarrheal samples, vs. 3.6 in Grif et al. This high prevalence is of concern regarding potential false positives, especially considering the threshold, which (at a million CFU/ml) is expected to be much higher than by culture. Selective enrichments such as by Sauders et al., and Grif et al. would actually have much lower threshold; and, if their assessments were based on colony, color etc. on the selective media, they would be more likely to over-estimate than under-estimate. These high prevalence levels in the current study were determined using a method (Le Monnier et al) developed for cerebrospinal fluid (CSF) from listeriosis patients, basically a monoculture for LM; however, human stool has immensely higher complexity, and the possibility of non-specific positive results cannot be excluded as potential contributor to the high observed prevalence in the current study. Another possibility might be that the LM detected in stools by PCR may have been inactivated subsequent to their passage through

the stomach and thus not recoverable by culture (but also not likely to respond to selection and contribute to dissemination).

(6) L228 “Lm persistence in food processing plants, away from its natural hosts, is associated with loss of virulence”. This statement can be misleading as stated. Food and food plant isolates of certain clonal groups tend to frequently harbor specific premature stop codons in one virulence gene, *inlA*, but others remain largely unaffected, as the authors themselves previously reported (Infect Immun. 2017 Oct 18;85(11):e00541-17. doi: 10.1128/IAI.00541-17). The available data suggest that the premature stop codons in *inlA* are more likely the result of positive selective pressure for adaptation to the food processing environment (as opposed to lack of purifying selection due to absence of “natural hosts”). Authors should also consider that LM from watersheds, also away from natural hosts, typically keep *inlA* and other virulence genes intact.

(7) Abstract: “All can thrive as saprophytes, whereas only pathogenic species cause systemic infections in human and cattle. “ Authors were suggested to replace human by “humans” in such uses (e.g. L37 etc., in many other locations) and they indicated in their Response that this was addressed but obviously the issue remains. The same applies with “16S” (L195, 466 etc).

(8) Abstract: “All can thrive as saprophytes, whereas only pathogenic species cause systemic infections in human and cattle. “yes, this is true, but as stated this statement can be misleading, as it omits the numerous other vertebrates that can become infected by LM. The manuscript does not get served well by such single-mined attention to the bovines. Authors may wish to consider “in humans and other vertebrates”.

(9) L64-66. Some large-scale surveys of LM in stool of healthy humans did include PCR, as authors also indicate later.

10. References vary in format, some use capitalized words, others not, bacterial names italics in some but not others, etc.

Reviewer #4 (Remarks to the Author):

Hafner et al. leverage community-level microbiome datasets and mouse models of *Listeria* carriage to demonstrate that *Listeria* fecal carriage is dictated by the microbiome. As reviewer 3 cannot review this version of the manuscript, I was requested to moderate the reviews from reviewer 3. As such, am focusing my review on whether or not reviewer 3’s initial concerns were appropriately addressed in the current manuscript. All comments not listed below have been adequately addressed, in my opinion.

INITIAL REVIEWER COMMENT

The mouse experiment shows different cage-batches of mice responding differently in terms of microbiome composition to a *Listeria* infection. The most important aspect of this experiment is to show that the microbiota prior to infection has an effect on the outcome. However this part relies on a small subset of mice (8 total), and the results are in supplemental (Fig S3). This needs to be replicated, it is not at all robust.

AUTHOR RESPONSE

We agree with the reviewer that the information provided in the previous Supplementary Fig. 3 is very valuable in supporting our conclusions. We have now performed additional experiments and have reinforced our findings and demonstrated the robustness of our conclusions.

To confirm the causal role of the microbiota on Lm carriage, we have reduced the alpha diversity, a key marker of a permissive microbiota in our mice and human datasets, by the use of a broad range antibiotic cocktail 4 weeks prior to infection and have compared carriage 30 dpi with PBS treated mice (new Figure 3f). While in antibiotic-treated mice 6/7 mice became Lm carrier ($>10^6$ CFU/g of faeces), none of the PBS-treated mice had any detectable Lm in the faeces at 30 dpi.

Further, we have characterized the change upon Lm infection in more mice (new Supplementary Figure 4a, b) and adapted our methodology to identify species that change upon Lm infection (DESeq2). We have removed claims that are not supported by the additional data, e.g. the decrease of Prevotellaceae upon infection.

Given that we have shown that i) the microbiota of carrier and non-carrier mice is different, ii) the inoculation with Lm does not alter the microbiota strongly (and hence cannot be responsible for the difference in i) and that iii) a decrease of the microbiota alpha diversity can induce a Lm carriage permissive state, we can conclude that the microbiota prior to Lm inoculation has an effect on asymptomatic Lm carriage, as

requested by the reviewer.

MY COMMENT

Reviewer 3 asked that the authors replicate the findings that the microbiota prior to infection has an effect on Lm carriage at 30 days, as the initial experiments performed which supported this conclusion leveraged a small number of mice classified as heavy carriers. The authors address this concern by performing a follow up experiment, where they pre-treat mice with an antibiotic cocktail or PBS and subsequently infect with Lm. They show that at 30dpi the antibiotic-treated animals have significantly more Lm detected in their feces relative to PBS-treated animals. They also show in Figure S4 that there are expected differences in diversity between antibiotic vs PBS treated controls. A few suggestions:

- Figure S4 should note that it's related to the experiment represented Figure 3F, as it's hard to tell based on the current figure cross-refs whether it refers to Figure 3A-3E or 3F.
- The color scheme in S4A should be revised as it's difficult to tell which points correspond to PBS treated versus antibiotic treated.
- The diversity metrics listed in S4C correspond to samples taken "4 weeks after treatment." Is this 4 weeks after antibiotic treatment or 4 weeks after Lm gavage (I assume 4 weeks after antibiotic treatment, immediately before Lm gavage, which is consistent with nomenclature through the rest of the paper)? Regardless, the authors should show these diversity metrics for both pre-infection and 30 days post infection in both Figures S4C and S2C if the conclusion is that diversity impacts the ability of mice to become Lm carriers and that Lm carriage influences diversity.

REVIEWERS' COMMENTS

We thank the reviewers for their helpful comments, which we have all addressed, both experimentally and by modifying the text.

Reviewer #1 (Remarks to the Author):

This paper presents the results of 3 of complementary studies addressing the relationship between prevalence of *Listeria monocytogenes* (LM) and the microbiota in (i) publicly available, previously sequenced environmental samples, (ii) among a cohort of 900 healthy and diarrheal human stool samples and (iii) an experimental mouse model. The paper is well-written, and the experiments and analysis of the data obtained human LM carriage and the experimental mouse model seem appropriate and do seem to support that (potentially) asymptomatic carriage of LM may be the factor driving purifying selection on virulence genes in this (mostly environmental) pathogen.

The authors have addressed my concerns/comments of the first version of this manuscript adequately, and I think the current revised manuscript is a great improvement, mainly in clarity of the methodology and thus reproducibility and transparency of the (bioinformatics) analyses.

Henk C. den Bakker

We thank reviewer #1 for his positive assessment of our revised manuscript.

Reviewer #2 (Remarks to the Author):

The manuscript has addressed several of the methodological issues brought up by the previous reviewers; however, concerns remain regarding certain methods and their potential outcomes, as well as regarding the interpretation of the findings.

We thank the reviewer for his/her assessment of our revised manuscript. Below and in the revised manuscript, we have addressed his/her remaining comments.

(1) The manuscript's key conclusion that "asymptomatic faecal carriage, rather than disease, exerts purifying selection on Lm "virulence genes" remains unsupported. Asymptomatic fecal carriage may indeed exert purifying selection, but this cannot be concluded from the current data, neither can it be concluded that it is the driving force.

We do not conclude that asymptomatic carriage is the driver behind this purifying selection, but only suggest this in the discussion of our results, since virulence associates with *Listeria ssp* fecal carriage.

And where, in this manuscript, is the genetic analysis of virulence genes during and following fecal carriage, to support purifying selection on these genes?

We do not conclude from our data that there is purifying selection on virulence genes, but this has been reported before (Maury *et al.*, 2017; Louie *et al.*, 2019) and is highlighted by the presence of most virulence genes in the *Lm* core genome, core genes being under purifying selection (Moura *et al.*, 2016) and their virtual absence in non-pathogenic *Listeria* species ((Orsi and Wiedmann, 2016; Moura *et al.*, 2019), see figure below). We propose in the discussion of the manuscript, that the asymptomatic carriage that associates with the presence

of virulence factors in *Listeria* ssp. could be the driver behind this previously reported purifying selection, but we would again like to underline that this is only a suggestion, not a conclusion.

Figure 1 Presence of virulence factors in genomes retrieved from bigsdb.pasteur.fr. Number of genomes per species shown on the left panel. In red the % of genomes that contain the corresponding virulence gene, in green *Imo0895*, which serves as a control for the quality of the genome and was present in all genomes used in the analysis.

Previous work with oral infection models (e.g., Roll and Czuprynski 1990, and Ref 6 by Louie et al) and mutants in virulence genes showed that upon oral inoculation LM could cause systemic infection only if it harbored intact virulence genes such as *hly* and *actA*. Ref 6 has shown that only LM with intact *hly* and *actA* could colonize the gallbladder (subsequent to liver colonization), from there seeding the intestine. But these were actual systemic infections, not asymptomatic carriage.

We agree with the reviewer that indeed invasion of host tissues is required for the establishment of *Lm* fecal carriage, given the known functions of the virulence factors that are present in *Lm* but absent in non-pathogenic species. The infectious dose we have used causes systemic infection with bacterial load in the liver and the spleen (e.g. (Maudet et al., 2020), unpublished data from the lab), we therefore consider that in our experimental setting systemic infection indeed preceded fecal carriage, which is detected at a time when no symptoms are present.

This means, that the purifying selection was still exerted at the level of disease (i.e., ability to colonize systemically, then colonize and grow in the gallbladder). In the current murine model which involved IV infection of the mice, LM likely entered the intestine following spillover from the liver to the gallbladder, i. e., following true systemic infection. Hence in this model as well, purifying selection is acting at the infection-disease level rather than at the level of fecal carriage (which is here taken as proxy for intestinal colonization, though this was not validated).

We failed to convey clearly what we understand under “asymptomatic carriage”, which is carriage without concomitant overt clinical disease, and is preceded by tissue invasion mediated by *Lm* virulence factors. We have modified the manuscript accordingly to take into account this important comment. This is a situation which is comparable to *Salmonella* enterica Typhi, which can be carried asymptotically in the feces at a time when no symptoms are observed (Gopinath, Carden and Monack, 2012)

(2) In the above context, even though “Asymptomatic” faecal carriage is being repeatedly mentioned, L171-6 also mentioned that “all mice recovered from infection as assessed by weight gain”; this suggests the presence of symptoms. Similarly, L208-9 indicates “Both antibiotic- and PBS-treated mice lost weight upon *Lm* inoculation”. The animal model system here does not support “asymptomatic carriage” because symptoms (e.g., weight loss), were observed.

We would like to insist again on what we call asymptomatic carriage, which is the presence of large quantities of *Lm* in the feces while concomitant symptoms are absent. We do not exclude symptoms before carriage is established (probably through systemic invasion and release from the gallbladder and possibly the intestinal tissue itself back to the intestinal lumen), but that they appear to be transient and benign in nature in the immunocompetent host compared to *Lm* carriage. This is highlighted by reports that even very high infectious doses ($>10^{11}$ CFU of *Lm*) only caused transient gastroenteritis and fever in the immunocompetent individuals (Dalton *et al.*, 1997; Aureli *et al.*, 2000).

(3) L170-8, were any assessments done on LM in liver, spleen or gallbladder in the mice? Also, any other symptoms besides weigh loss? Authors should also elaborate as to whether the weight loss was accompanied with fever and lack of appetite.

The inoculation indeed leads to transient weight loss in some animals, but we assessed the animals daily and they did not display any sign of disease. We have changed the text and mention this now. In a subset of carrier mice (n=4), the CFU count in inner organs was assessed at day 30 and no CFU were detected (data not shown).

(4) Authors, consider clarifying why an IV model was employed to assess intestinal colonization, instead of available oral inoculation models such as mentioned above. An oral model would be clearly advantageous for investigating the host and bacterial outcomes of ingestion of LM via foods. It would require suppression of the host interference with the gut microbiota, e.g., via antibiotic treatment, but antibiotic treatments were also used in the current model to make the gut more permissive to LM.

We used an *iv* model because the interactions between the gut microbiota and the incoming *Lm* would not allow to independently assess its impact on fecal carriage, which could potentially overshadow the effect of different microbiota compositions on *Lm* carriage, e.g. by the action of bacteriocins on the gut microbiota (Quereda *et al.*, 2016) and the impact of colonization resistance on tissue invasion (Becattini *et al.*, 2017). We have clarified the reasoning behind *iv* inoculation in the text now.

(5) The PCR-based defection of LM in the human stool samples remains problematic. Even though in their Response the authors indicate that the PCR prevalence was similar in the current study and Grif *et al.*, the data in the manuscript indicate 10% in healthy humans and even twice that in diarrheal samples, vs. 3.6 in Grif *et al.* This high prevalence is of concern regarding potential false positives, especially considering the threshold, which (at a million CFU/ml) is expected to be much higher than by culture. Selective enrichments such as by Sauders *et al.*, and Grif *et al.* would actually have much lower threshold; and, if their assessments were based on colony, color etc. on the selective media, they would be more likely to over-estimate than under-estimate.

These high prevalence levels in the current study were determined using a method (Le Monnier *et al.*) developed for cerebrospinal fluid (CSF) from listeriosis patients, basically a monoculture for LM; however, human stool has immensely higher complexity, and the possibility of non-specific positive results cannot be excluded as potential contributor to the high observed prevalence in the current study.

To ensure that the approach of Le Monnier *et al.* did not produce any false positive results due to the complex environment that are the feces, we have now amplified and sequenced another region of *hly* in 10 positives fecal samples (position 350 to 820) and found all of them to correspond to the sequence of *hly* (>97.5% identity). We have added this to the results as Supplementary File 1 and mention it in the methods section.

Another possibility might be that the LM detected in stools by PCR may have been inactivated subsequent to their passage through the stomach and thus not recoverable by culture (but also not likely to respond to selection and contribute to dissemination).

We agree with the reviewer that we cannot distinguish between viable and dead *Lm*. We mention this now in the text.

(6) L228 “*Lm* persistence in food processing plants, away from its natural hosts, is associated with loss of virulence”. This statement can be misleading as stated. Food and food plant isolates of certain clonal groups tend to frequently harbor specific premature stop codons in one virulence gene, *inlA*, but others remain largely unaffected, as the authors themselves previously reported (Infect Immun. 2017 Oct 18;85(11):e00541-17. doi: 10.1128/IAI.00541-17). The available data suggest that the premature stop codons in *inlA* are more likely the result of positive selective pressure for adaptation to the food processing environment (as opposed to lack of purifying selection due to absence of “natural hosts”).

The reviewer raises a very interesting point, since it is indeed unclear if the premature stop codon is an adaptive mutation, the result of the release of purifying selection, or both. Nevertheless, it is clear that these isolates are less virulent and that this loss of virulence is recent (Maury *et al.*, 2016, 2019; Moura *et al.*, 2016). Further, isolates carrying a premature stop codon in *inlA* colonize less the mammalian gut (Maury *et al.*, 2019).

Authors should also consider that LM from watersheds, also away from natural hosts, typically keep *inlA* and other virulence genes intact.

We agree with the reviewer that most environmental isolates have intact virulence factors, but we cannot exclude that these bacteria frequently transit *via* a host environment where these factors have a function, contrary to isolates persisting in food production plants, where no potential host is present.

To address the reviewer’s comment, we have edited the text, to make it clearer that we do not conclude on purifying selection and that future research will need to evaluate the relative importance of each environment for this selection. Also, we have omitted the word “natural” host, as we cannot assess definitely which is the natural host of *Lm*, if there is any.

(7) Abstract: “All can thrive as saprophytes, whereas only pathogenic species cause systemic infections in human and cattle. “ Authors were suggested to replace human by “humans” in such uses (e.g. L37 etc., in many other locations) and they indicated in their Response that this was addressed but obviously the issue remains. The same applies with “16S” (L195, 466 etc).

We have changed the outlined sections.

(8) Abstract: “All can thrive as saprophytes, whereas only pathogenic species cause systemic infections in human and cattle. “yes, this is true, but as stated this statement can be misleading, as it omits the numerous other vertebrates that can become infected by LM. The manuscript does not get served well by such single-mined attention to the bovines. Authors may wish to consider “in humans and other vertebrates”.

We have changed the outlined section.

(9) L64-66. Some large-scale surveys of LM in stool of healthy humans did include PCR, as authors also indicate later.

We have changed the text and mention that these studies “mostly” applied culture-based methods.

10. References vary in format, some use capitalized words, others not, bacterial names italics in some but not others, etc.

We have re-formatted the references.

Reviewer #4 (Remarks to the Author):

Hafner et al. leverage community-level microbiome datasets and mouse models of *Listeria* carriage to demonstrate that *Listeria* fecal carriage is dictated by the microbiome. As reviewer 3 cannot review this version of the manuscript, I was requested to moderate the reviews from reviewer 3. As such, am focusing my review on whether or not reviewer 3's initial concerns were appropriately addressed in the current manuscript. All comments not listed below have been adequately addressed, in my opinion.

INITIAL REVIEWER COMMENT

The mouse experiment shows different cage-batches of mice responding differently in terms of microbiome composition to a *Listeria* infection. The most important aspect of this experiment is to show that the microbiota prior to infection has an effect on the outcome. However this part relies on a small subset of mice (8 total), and the results are in supplemental (Fig S3). This needs to be replicated, it is not at all robust.

AUTHOR RESPONSE

We agree with the reviewer that the information provided in the previous Supplementary Fig. 3 is very valuable in supporting our conclusions. We have now performed additional experiments and have reinforced our findings and demonstrated the robustness of our conclusions.

To confirm the causal role of the microbiota on *Lm* carriage, we have reduced the alpha diversity, a key marker of a permissive microbiota in our mice and human datasets, by the use of a broad range antibiotic cocktail 4 weeks prior to infection and have compared carriage 30 dpi with PBS treated mice (new Figure 3f). While in antibiotic-treated mice 6/7 mice became *Lm* carrier ($>10^6$ CFU/g of faeces), none of the PBS-treated mice had any detectable *Lm* in the faeces at 30 dpi. Further, we have characterized the change upon *Lm* infection in more mice (new Supplementary Figure 4a, b) and adapted our methodology to identify species that change upon *Lm* infection (DESeq2). We have removed claims that are not supported by the additional data, e.g. the decrease of *Prevotellaceae* upon infection. Given that we have shown that i) the microbiota of carrier and non-carrier mice is different, ii) the inoculation with *Lm* does not alter the microbiota strongly (and hence cannot be responsible for the difference in i and that iii) a decrease of the microbiota alpha diversity can induce a *Lm* carriage permissive state, we can conclude that the microbiota prior to *Lm* inoculation has an effect on asymptomatic *Lm* carriage, as requested by the reviewer.

MY COMMENT

Reviewer 3 asked that the authors replicate the findings that the microbiota prior to infection has an effect on *Lm* carriage at 30 days, as the initial experiments performed which supported

this conclusion leveraged a small number of mice classified as heavy carriers. The authors address this concern by performing a follow up experiment, where they pre-treat mice with an antibiotic cocktail or PBS and subsequently infect with *Lm*. They show that at 30dpi the antibiotic-treated animals have significantly more *Lm* detected in their feces relative to PBS-treated animals. They also show in Figure S4 that there are expected differences in diversity between antibiotic vs PBS treated controls. A few suggestions:

We would like to thank the reviewer #4 for the positive assessment of our revised manuscript.

- Figure S4 should note that it's related to the experiment represented Figure 3F, as it's hard to tell based on the current figure cross-refs whether it refers to Figure 3A-3E or 3F.

We have changed subtitle of Figure S4 to make clear that it refers to Figure 3F.

- The color scheme in S4A should be revised as it's difficult to tell which points correspond to PBS treated versus antibiotic treated.

We have changed the color scheme of figure S4A and other figures where we distinguish between antibiotic treated and PBS control mice.

- The diversity metrics listed in S4C correspond to samples taken "4 weeks after treatment." Is this 4 weeks after antibiotic treatment or 4 weeks after *Lm* gavage (I assume 4 weeks after antibiotic treatment, immediately before *Lm* gavage, which is consistent with nomenclature through the rest of the paper)? Regardless, the authors should show these diversity metrics for both pre-infection and 30 days post infection in both Figures S4C and S2C if the conclusion is that diversity impacts the ability of mice to become *Lm* carriers and that *Lm* carriage influences diversity.

The reviewer is right that the displayed data corresponds to 4 weeks after the antibiotic treatment. We have added the 4 weeks after *Lm* inoculation to increase the clarity of the figure. Of note, our conclusion nevertheless is not "that diversity impacts the ability of mice to become *Lm* carriers and that *Lm* carriage influences diversity", but that mainly diversity impacts carriage.

REVIEWERS' COMMENTS

Reviewer #2 (Remarks to the Author):

The revision has addressed several of the previously-raised issues. Two comments:

1, Authors indicate in their RESPONSE file: "We do not conclude that asymptomatic carriage is the driver behind this purifying selection, but only suggest this in the discussion of our results, since virulence associates with *Listeria ssp* fecal carriage."

Yet, The Abstract concludes by "These results suggest that faecal carriage, rather than disease itself, exerts purifying selection on *Lm* "virulence genes"."

The authors do use "suggest", but the thrust of this concluding statement is in contrast with the above statement in the authors' RESPONSE file, and remains largely unsupported by the findings. Furthermore, how can this be concluded when fecal carriage is actually preceded by disease? Is it reasonable to expect that mutations leading to loss or attenuation of virulence would accrue and become detectable, with the methods that were employed, in just 27 days, i.e., the period between cessation of symptoms and the fecal carriage assessments? Previous studies have shown that feces become colonized via bacteria from the gallbladder, and that to colonize the gallbladder *Lm* needs to be virulent.

L226-229. besides cattle several other vertebrates, bath domesticated animals and wildlife, have been found to commonly shed LM in the feces.

Unité de Biologie des Infections
Inserm U1117

Paris, November 29, 2021

Reviewer #2 (Remarks to the Author):

The revision has addressed several of the previously-raised issues. Two comments:

1, Authors indicate in their RESPONSE file: “We do not conclude that asymptomatic carriage is the driver behind this purifying selection, but only suggest this in the discussion of our results, since virulence associates with *Listeria ssp* fecal carriage.”

Yet, The Abstract concludes by “These results suggest that faecal carriage, rather than disease itself, exerts purifying selection on *Lm* “virulence genes”.”

The authors do use "suggest", but the thrust of this concluding statement is in contrast with the above statement in the authors' RESPONSE file, and remains largely unsupported by the findings.

Furthermore, how can this be concluded when fecal carriage is actually preceded by disease? Is it reasonable to expect that mutations leading to loss or attenuation of virulence would accrue and become detectable, with the methods that were employed, in just 27 days, i.e., the period between cessation of symptoms and the fecal carriage assessments? Previous studies have shown that feces become colonized via bacteria from the gallbladder, and that to colonize the gallbladder *Lm* needs to be virulent.

We have edited the manuscript to address this point.

L226-229. besides cattle several other vertebrates, bath domesticated animals and wildlife, have been found to commonly shed LM in the feces.

We mention on L286 that other vertebrates also shed *Lm*. We have modified the sentence to make it clear that we do not provide an exhaustive list of species in which asymptomatic *Lm* carriage has been reported.